# A Pictorial Review of the Role of Imaging in the Detection, Management, Histopathological Correlations, and Complications of COVID-19 Pneumonia

**DOI:** 10.3390/diagnostics11030437

**Published:** 2021-03-04

**Authors:** Barbara Brogna, Elio Bignardi, Claudia Brogna, Mena Volpe, Giulio Lombardi, Alessandro Rosa, Giuliano Gagliardi, Pietro Fabio Maurizio Capasso, Enzo Gravino, Francesca Maio, Francesco Pane, Valentina Picariello, Marcella Buono, Lorenzo Colucci, Lanfranco Aquilino Musto

**Affiliations:** 1Department of Radiology, San Giuseppe Moscati Hospital, Contrada Amoretta, 83100 Avellino, Italy; menavolpe@hotmail.com (M.V.); giuliolombardi1@gmail.com (G.L.); segreteria.alessandrorosa@gmail.com (A.R.); giuliano.gagliardi@hotmail.it (G.G.); p.capasso4@virgilio.it (P.F.M.C.); enzogravino@hotmail.it (E.G.); francescomaio9@gmail.com (F.M.); dr.panef@gmail.com (F.P.); vntpink@gmail.com (V.P.); marcella.buono@alice.it (M.B.); lorenzocolucci@inwind.it (L.C.); musto.lanfranco@gmail.com (L.A.M.); 2Radiology Unit, Cotugno Hospital, Naples, Via Quagliariello 54, 80131 Naples, Italy; dr.eliobignardi@alice.it; 3Neuropsychiatric Unit ASL Avellino, Via Degli Imbimbo 10/12, 83100 Avellino, Italy; claudiabrogna@yahoo.it

**Keywords:** COVID-19, COVID-19 pneumonia imaging guidelines, chest X-ray, chest X-ray protocols, chest X-ray scoring system, lung ultrasound, lung ultrasound protocols, lung ultrasound scoring system, chest CT, chest CT protocols, chest CT severity scores, ARDS, histopathological correlations, COVID-19 complications

## Abstract

Imaging plays an important role in the detection of coronavirus (COVID-19) pneumonia in both managing the disease and evaluating the complications. Imaging with chest computed tomography (CT) can also have a potential predictive and prognostic role in COVID-19 patient outcomes. The aim of this pictorial review is to describe the role of imaging with chest X-ray (CXR), lung ultrasound (LUS), and CT in the diagnosis and management of COVID-19 pneumonia, the current indications, the scores proposed for each modality, the advantages/limitations of each modality and their role in detecting complications, and the histopathological correlations.

## 1. Introduction

The coronavirus (COVID-19) pandemic, caused by a novel coronavirus named severe acute respiratory syndrome coronavirus-2 (SARS-CoV-2), continues to put stress on health care around the world and trouble the economy of a lot of countries. Although COVID-19 was officially recognized in Wuhan, Hubei Province, China, in December 2019, the epidemic could have started much earlier [1,2]. COVID-19 can be considered a multi-organ disease with heterogeneous manifestations, which vary from asymptomatic to pneumonia, gastrointestinal and neurological presentations, and acute respiratory distress syndrome (ARDS) [3,4,5,6,7]. Early anosmia and dysgeusia may be present [3,6]. SARS-CoV-2 is usually transmitted by respiratory droplets and fomites. However, there is also evidence of fecal–oral transmission [5,7]. Although COVID-19 can manifest as a multi-organ disease, the lung is considered a preferential site by the virus.

To date, reverse-transcription polymerase chain reaction (RT-PCR) testing is considered to be the gold standard tool to diagnose or screen for COVID-19. Despite its high specificity, RT-PCR sensitivity ranges from 37 to 71% [8,9], with false negative results reported in 50% of cases [10]. The limited testing capacity can be due to low or insufficient viral load in the specimen collection or the sample collection time. Thoracic imaging with chest radiography (chest X-ray, CXR) and computed tomography (CT) is usually considered a key tool to support pulmonary disease diagnosis and management. Chest imaging is currently indicated in COVID-19 patients with acute respiratory illness and in suspected cases to support rapid medical triage in the presence of high pretest probability for patients at high-moderate risk of progression [11,12]. Although CXR is usually suggested as the first imaging tool for suspected cases with respiratory symptoms in the emergency department (ED) [13,14,15,16,17,18], the choice of imaging modality between CXR and CT is usually left to the judgment of the clinical team, the availability of local resources, and the expertise of radiologists [11,12,18]. CT is mainly useful in suspected cases of COVID-19 pneumonia with the presence of comorbidities when there is a high risk of complications [19,20]. However, CT can also allow management with the isolation of suspected cases in the presence of typical imaging features pending swab RT-PCR results [21]. The role of lung ultrasound (LUS) remains controversial. Therefore, the major radiology societies [11,12,13,14,15,16,17,18,19,20,22,23] do not currently recommend the use of imaging for screening or as a unique tool to diagnose COVID-19. The main international guidelines are summarized in Table 1. Confirmation with RT-PCR is always required even in the presence of typical imaging findings [21,22]. However, chest imaging has an important role in diagnostic support in the presence of typical imaging findings for COVID-19 and multiple negative RT-PCR results in the presence of high pretest probability [11,12,24]. In these selective cases, confirmation should be conducted with serology along with laboratory examinations [24]

## 2. Chest X-ray

### 2.1. Chest X-ray: Role and Limitations

Chest computed tomography (CT) and chest radiography (chest X-ray, CXR) are the imaging modalities generally performed to detect lung abnormalities in early infection, and to assess the severity and monitor the progression of COVID-19 pneumonia [25,26]. CT has been mainly used for diagnosing COVID-19 pneumonia [27,28,29]. This is due to the experience of China, where access to CT was relatively easy at the outbreak of the pandemic [28], and the low sensitivity of CXR in revealing pulmonary involvement, particularly in the early stage of the disease [25,27]. Using RT-PCR as the gold standard, the sensitivity of CRX, which depends not only on the disease stage, but also on its severity, the prevalence of COVID-19 in the area, and the expertise of radiologists, with possible support of artificial intelligence (AI), differs widely from 41.7 to 90% and is still lower than CT (60 to 98%) [30,31,32]. The specificity, on the other hand, is between 33 and 60.6% [33,34]. To date, there have been few studies on the diagnostic accuracy of CXR, and their significance is limited by small sample sizes, often without the inclusion of healthy individuals or non-COVID-19 cases [33,35].

In the emergency care setting, especially in biocontainment units (BUs) and intensive care units (ICUs), and in critically ill patients, the role of CXR is relevant [15,27,35]. The European Society of Radiology (ESR) and the European Society of Thoracic Imaging (ESTI) suggested CXR for follow-up of patients admitted to the ICU who are too fragile to be sent to CT [19]. On the other hand, many radiology societies suggest CXR as the first modality of choice for the diagnostic work-up of COVID-19 patients [13,14,15,16,17,18]. However, it should also be considered that the imaging modality to use usually depends on the judgment of clinical teams, the availability of local resources, and the expertise of radiologists. In fact, the use of chest CT was dominant in China and Italy at the beginning of the pandemic [21,36].

According to recent literature data, CXR shows radiographic features in most patients with COVID-19, including reticular alteration, ground-glass opacity (GGO), consolidations, bilateral involvement, peripheral distribution, lower zone dominance, pleural effusion, and reduction in lung volume, as well as complications such as pneumothorax (PNX), pneumomediastinum (PM), and subcutaneous emphysema, consistent with previous reports on CT [8,35,37]. Furthermore, the effective dose (mSv) is lower for CXR than CT [38]. Therefore, some limitations including higher ionizing radiation exposure for CT than CRX, as well as the restricted availability of dedicated CT scanners, difficulties in speeding up CT room sanitization, the risk of transmitting SARS-CoV2 infection during patient transfer to the CT room, and the complexity of transporting unstable patients reduce the predominance of CT over CXR [11,23,25,35]. Moreover, the rise in radiological examinations that comes with the increase in hospitalized patients makes the continuous use of chest CT throughout the course of the disease problematic [26]. In addition, the American College of Radiology (ACR) noted that CT room decontamination after scanning patients with COVID-19 may disrupt the availability of radiological services and suggested that portable CXR might be considered the optimal tool to minimize the risk of cross-infection [23]. Therefore, European hospitals, particularly in Italy and Great Britain (Table 1), are beginning to employ CXR (bedside or standard) as the first radiological modality in patients presenting with respiratory distress and possible COVID-19 and for monitoring the evolution of lung abnormalities, particularly in critically ill patients admitted to the ICU, where CXR can also be used to evaluate chest tube positioning (Figure 1) [12,13,23,25,27,35].

### 2.2. Chest X-ray: Protocols

CXR is a low-cost and widely available tool. Different protocols have been proposed for the use of CXR in radiology departments (RDs) and at the bedside in hospital wards and BUs, and in emergency departments (EDs) [35,39]. All CXRs are acquired as computed or digital radiographs [27,30,35,40]. In the current pandemic context, CXR protocols in isolation rooms in EDs are particularly significant. In order to prevent the risk of transmitting SARS-CoV2 infection, as for Ebola virus disease (EBD), definitions have been provided for hot, cold, and transition zones in BUs; compliance with dressing and undressing procedures for the operators; and decontamination of the environment and equipment [39,41]. The procedures involve two dedicated operators (usually two radiographers) [39,41,42]. Personal protective equipment (PPE) should be worn, and the level of protection will vary according to the tasks that need to be performed [39,41]. The highest level of protection, the third level, is for operators who will be in the same room as a patient suspected of having or confirmed to have SARS-CoV2 infection [39,41,42]. The third level of protection requires the use of a full barrier, which is a combination of airborne and contact precautions, plus eye protection, in addition to standard precautions [39,41,42]. CXRs that are obtained at the bedside, even in critically ill patients in isolation rooms, are acquired in the supine position, usually antero-posterior (AP) projection only, using a portable CXR unit [35,39]. The recommended source–detector distance is 100 cm, and the exposure parameters are 75–85 KV with an X-ray grid and 65–70 KV without an X-ray grid [35]. The radiographic cassettes must be contained in a double- or triple-sealed fluid-proof plastic bag or a plastic cover and a clean pillowcase and properly disinfected at the end of the procedure [35]. The images are stored in a picture archiving and communication system (PACS) [30,37].

### 2.3. Chest X-ray: Scoring System

To assess COVID pneumonia and monitor its progression, several CXR scoring systems and structured reports have been developed [27,35,43,44,45,46,47]. In 2015, Taylor et al. [27,43] proposed a scoring system designed for non-radiologist clinicians that could be applied to COVID pneumonia in order to facilitate clinical grading in five severity categories of CXR reports for hospitalized patients with acute respiratory infection [27,43]. Maroldi et al. [27] presented a new experimental CXR scoring system, named the Brixia score, for semi-quantitative assessment of COVID-19 pneumonia applicable to hospitalized patients with infection confirmed by RT-PCR. In the first step, the lungs are divided into six zones on a postero-anterior (PA) or antero-posterior (AP) projection. In the second step, a score (0 to 3) is assigned to each zone based on lung abnormalities (0 = no lung abnormalities; 1 = interstitial infiltrates; 2 = interstitial and alveolar infiltrates with interstitial predominance; 3 = interstitial and alveolar infiltrates with alveolar predominance). The scores of the six lung zones are then added to obtain an overall CXR score ranging from 0 to 18 (Figure 2) [27].

Cozzi et al. [47] applied the radiographic assessment of lung edema (RALE) score, proposed by Warren et al. [44], to COVID-19 pneumonia: scores range between 0 (absence of pathology) to 48 (total pathological involvement of lung) [47]. Wong et al. [35] simplified and adapted the RALE score: a score of 0–4 is assigned to each lung depending on the extent of involvement with consolidation or GGO (0 = no involvement; 1 = <25%; 2 = 25–50%; 3 = 50–75%; 4 = >75% involvement) [35]. The scores for each lung are summed to produce the final severity score, ranging from 0 to 8 [35]. Toussie et al. [45] studied the relationship between clinical and initial CXR findings and the outcome variables of hospital admission and/or intubation in COVID-19 patients between the ages of 21 and 50 years. They divided each lung into three zones, and each zone is given a binary score depending on whether opacity is absent (0) or present (1); total scores range from 0 to 6 [45]. A higher disease score at baseline would be associated with a requirement for mechanical ventilation and increased risk of ICU admission and in-hospital mortality [45,47]. Finally, some recent literature data suggest that AI systems can improve performance in evaluating CXRs, including activation of parametric maps, and in the future possibly by supporting radiologists in quantifying COVID-19 pneumonia [30,48].

## 3. Lung Ultrasound

### 3.1. Lung Ultrasound: Role and Limitations

While chest computed tomography (CT) is considered the gold standard imaging procedure in COVID-19 pneumonia [27,28,29], difficulties in transporting critically ill patients, the risk of cross-infection during patient transfer to imaging departments, required decontamination after scanning patients, and high exposure to ionizing radiation limit its role [11,23,25,33,35]. On the other hand, LUS has some advantages as a portable and radiation-free tool. LUS is a low-cost imaging technique that is quick to perform and repeat at the bedside by emergency physicians, intensivists, or cardiologists (point-of-care (POC) LUS). This technique allows a dynamic study of the lung without ionizing radiation exposure and with a reduced overall nosocomial transmission risk, also useful in vulnerable groups such as children and pregnant women [49,50,51,52,53,54,55]. LUS can detect bilateral, subpleural, mainly posterobasal interstitial–alveolar damage in COVID-19 pneumonia with the appearance of thickening/irregularity of the pleural line, increased B lines to different degrees of extension with focal pleural B lines in the early stage of the disease, and multiple coalescent B lines (white lung) in critically ill patients [53,54,56]. LUS can also detect small multifocal consolidations adherent to the subpleural surface, as well as pleural effusions [53,54,55]. However, LUS cannot accurately detect the presence of air bronchogram as well as pneumothorax [57,58]. Some authors have noted that LUS may have a potential role in EDs for triaging symptomatic patients, managing ventilation, weaning ICU patients, and monitoring COVID-19 pneumonia and its evolution toward ARDS in critically ill patients [49,50,51,55,59,60,61,62,63,64,65]. A focused cardiac ultrasound study (FoCUS) performed at the bedside can also be useful in COVID-19 patients with cardiac events [64]. Therefore, LUS may be considered as a first-line alternative to chest X-ray and CT scan in critically ill patients [54,59,60,61,62,63,66]. However, despite the high sensitivity and diagnostic accuracy reported for LUS, the specificity is low, ranging between 59 and 76.2% [49,65]. Although LUS is a promising tool, an international consensus has not been reached on its use in the management of COVID-19. This may be due to its many limitations. First, LUS is constantly limited by the presence of artifacts related to air in the lungs; thus, it is mainly based on the interpretation of imaging artifacts [67,68,69,70,71,72]. Two main artifacts are created by the physical interaction of the ultrasonic beam with the tissue–air interface: the horizontal A line and vertical B line (Figure 3) [67,68]. B lines are usually found in the pleuropulmonary physiology or various pathological conditions in which the air–liquid interface is altered [71,72,73,74,75,76,77,78]. Therefore, B lines are not specific markers of interstitial edema. Multiple B lines can also be found in other pulmonary diseases, such as pulmonary edema due to cardiovascular disease, aspiration, ARDS, interstitial lung disease, or other pneumonias (Figure 3) [66,71,72,75,76,77,78,79,80]. According to Sperandeo et al. [71], the specificity of LUS in COVID-19 patients is usually low since these patients have co-morbidities such as chronic obstructive pulmonary disease, fibrosis, or heart failure. Compared with volumetric chest CT, LUS explores, at best, with the patient in a sitting position, only 70% of the pleural surface and does not identify the central regions as well as the perihilar or subpleural regions that do not adhere to the pleural surface, due to the interposition of a very thin layer (micron or mm) of air between the pleural surface surrounded by the aerated lung [73,76,77]. Instead, LUS can be effectively used for exploring lesions or conditions confined to the pleural or subpleural regions, such as in the evaluation of pleural effusions, subpleural lesions, the chest wall, and the upper anterior mediastinum. On the other hand, LUS is strongly operator-dependent [71,72,77]. The risk that LUS will be ineffective in untrained hands may be more harmful than helpful. LUS is generally based on subjective observations. Pleural line abnormalities may vary based on the type of probe used, the angle of incidence of the probe, the ultrasound scan (longitudinal, transverse, or oblique), and the operator’s experience [77,80]. Quarato et al. [76,77,80] and Sperandeo et al. [71,75] highlighted the technical limits of LUS. In particular, the intra- and inter-operator variability in B line counting depends on the type and frequency of the probe used and the ultrasound scan machine setting [71,75,76,77,79]. Therefore, the use of medium to low frequency or excessive total gain and the lack of tissue harmonic imaging can generate a larger number of artifacts [77,79]. Tinti et al. [74] pointed out that in order to obtain real and valid quantification of B lines, the physician should freeze the ultrasound image and count the lines every time the probe position changes. In this context, Carrer et al. [59] suggested an automatic method for detecting pleural lines. Finally, despite the use of full PPE and rigorous observance of decontamination procedures, operators performing LUS may be placed at increased risk of contracting COVID-19 [39,81].

### 3.2. Lung Ultrasound: Protocols

The most commonly proposed protocols are the bedside lung ultrasonography in emergency (BLUE), fluid administration limited by lung sonography (FALLS), and cardiac arrest ultrasound exam (CAUSE) protocols [82]. Most LUS scans are performed at the bedside, initially with the patient in a sitting position, because this is the best posture for a patient experiencing dyspnea, and then in a supine position, to conduct a better diaphragm evaluation [67,74]. However, in the emergency setting, LUS can be performed in any position that is comfortable for the patient [67]. The physician should use both a multifrequency 3–8 MHz convex probe and a high-frequency linear probe (8–12.5 MHz) [50,67]. The probe depth should range from 70 to 140. The use of tissue harmonics is preferable to reduce natural artifacts, with a time gain compensation (TGC) not exceeding 55% [67]. The examination includes the exploration of 6, 8, 12, 14, or 28 zones [51,83,84,85,86,87] (Figure 4). The duration depends on the number of areas being examined, which can be less than 2 min if six lung zones are examined, as in the focused ultrasound in intensive care (FUSIC) lung accreditation module [85]. The use of PPE and adherence to decontamination procedures must be rigorous. The World Federation for Ultrasound in Medicine and Biology Safety Committee provides guidance on cleaning equipment and safe performance of ultrasound examinations within the context of COVID-19 [81]. Generally, physicians should use all necessary PPE and follow appropriate and complete dressing procedures. The ultrasound probe must be cleaned with appropriate sprays or disinfectants before and after use and be covered with a plastic film [77].

### 3.3. Lung Ultrasound Scoring System

Although LUS is not currently considered in the main international guidelines for COVID-19 patient management, some authors have proposed semi-quantitative LUS scores for COVID-19 pneumonia, which can be used to quantify lung aeration [55,61,82,83,84,85,86,87,88,89,90,91,92,93]. The most widely used is called the lung ultrasound score (LUSS) [61], global or total LUS score [89,91], or global LUS aeration score [93], in which 0–3 points are assigned for each of the 12 zones according to ultrasound features [84,85,86,87,89,90]: 0 = normal; 1 = well-defined B lines (B1); 2 = coalescent B lines or white lung (B2); and 3 = consolidation (Figure 5). Total scores range from 0 (best) to 36 (worst) [90,91]. Severity is classified as mild (1–5), moderate (>5–15), and severe (>15) [84]. A correlation between total LUS score and CT severity score (CT-SS) of 0 to 20 is good [86]. Deng et al. [83] presented another score based on examination of 8 zones, ranging from 0 (best) to 24 (worst), and correlating well with a CT severity score (CT-SS) of 0 to 25. Peschel et al. [93] used 0 to 4 points for up to 12 zones; this score, called the lung aeration score (LAS), is assessed by calculating the arithmetic mean of the points of all examined areas. Recently, Zotzmann et al. [92] advocated a combination of LUS and the Wells score as a screening tool for pulmonary embolism (PE) in COVID-19 ARDS. The baseline LUS score (within 24 h of admission) would be roughly linked to the eventual need for invasive mechanical ventilation and would be a strong predictor of mortality [91]. Finally, automatic pleural line extraction and a predicted score chart can provide an automatic COVID-19 score from LUS data [59].

## 4. Chest CT

### 4.1. Chest CT: Role and Limitations

Chest CT, due to its high sensitivity and widespread availability, is considered to be an effective tool to support the diagnosis of COVID-19 pneumonia with high or intermediate clinical probability, since it allows a rapid diagnostic workflow in emergency triage and may partially overcome the long turnaround time of RT-PCR [11,94,95,96,97,98,99]. In the emergency setting, patients with CT findings typical of COVID-19 pneumonia can be promptly isolated. However, a study by Skalidis et al. [97] showed that CT did not modify the estimated probability of COVID-19 infection and RT-PCR testing is generally required to confirm the diagnosis [11,12,13,14,15,16,17,18,19,22,23]. With the use of RT-PCR as the gold standard tool, chest CT usually showed greater than 90% sensitivity, with low to moderate specificity values ranging between 25 and 56% [8,100]. A recent meta-analysis by Kim et al. [101] reported a pooled sensitivity for chest CT of 94%, with a pooled specificity of 37%. Compared with CXR, chest CT showed increased sensitivity for COVID-19 pneumonia, with improved diagnostic performance [102]. Borakati et al. [102] reported no difference in specificity between the two modalities.

The sensitivity of chest CT usually depends on the phase of the disease. Some studies reported that CT sensitivity can be higher at initial presentation of the disease compared to RT-PCR [103,104,105,106]. False negative RT-PCR tests have been also reported for patients with CT findings of COVID-19 who eventually tested positive with serial sampling [104,105,106,107]. CT sensitivity in a study by Guillo et al. [108] increased after 5 days of symptoms. Pan et al. [109] found that lung alterations showed the greatest extension on CT approximately 10 days after the onset of symptoms. Although there was low CT specificity when a “reverse calculation” approach was used, chest CT could have higher specificity (83–100%) [110].

CT plays a relevant role in supporting COVID-19 diagnosis in symptomatic patients in the presence of multiple repetitive negative RT-PCR results [24,111]. However, patients with RT-PCR-confirmed COVID-19 may also have normal CT findings, especially in the first 3 days after the onset of symptoms [28,104]. Chest CT can also reveal lung alterations in asymptomatic or mildly symptomatic patients [112,113], but its use as a screening tool is not validated due to the amount of radiation and the possibility of normal findings [11,12,13,14,15,16,17,18,19,20,21,22,23,114]. CT is also used during short-term follow-up of COVID-19 for clinical decision making to monitor the course of pneumonia in hospitalized patients, and it can also be used in symptomatic patients for long-term follow-up, as it can reveal late lung alterations [29,108,113,114,115,116]. Fu et al. [117] suggested that CT was better than nucleic acid conversion in assessing final treatment outcomes. The diagnostic value of CT decreases in cases of low prevalence of COVID-19 disease [101]. In this setting, chest CT findings can overlap with other diseases such as other viral pneumonia, organizing pneumonia (OP), drug toxicity, and pulmonary edema [99,118]. Although chest CT features are not specific, experienced radiologists can diagnose the disease [99,119,120]. In the future, increased use of artificial intelligence could help to overcome this limitation [121]. The major radiology societies [11,12,13,14,15,16,17,18,19] recommend the use of CT in the presence of moderate and severe features of COVID-19 when RT-PCR results are negative or not available, when there is high pre-test probability, and in the management of patients with worsening or severe respiratory symptoms.

### 4.2. Chest CT Protocols

High-resolution chest CT scans for COVID-19 patients should be performed with volumetric acquisitions in deep inspiration with a slice thickness <3 mm [122,123]. Low-dose protocols with lower kilovoltage settings and iterative or deep learning-based reconstruction are preferred in order to minimize the radiation burden because COVID-19 patients can undergo various CT examinations during follow-up [121,122,123,124,125,126]. Expiratory phase CT does not add diagnostic value and can increase the radiation dose. Post-contrast CT angiography, with contrast material injected at a high rate (>3 mL/s) and bolus tracking techniques, is usually performed when pulmonary embolism is suspected (Figure 6) [127]. When pneumonia is already known, a CT angiograph can be directly acquired [127]. Perfusion lung abnormalities can be detected with dual-energy CT angiography [128]. Radiology personnel should use appropriate PPE, including face masks, eye protection, and gloves, with correct donning and doffing procedures in a dedicated and separate room. Deep cleaning of the CT examination room with room downtime of 30 min to 1 h, and increased air exchange or high-efficiency particulate air (HEPA) filtration are necessary [11,39,122,129].

### 4.3. COVID-19 CT Features and Reporting System

On CT, it is possible to recognize typical patterns of COVID-19 pneumonia that are usually those more frequently reported in the recent literature according to the Fleischner Society nomenclature [130], varying from bilateral pulmonary parenchymal GGO to consolidations, reticular interlobular septal thickness, crazy paving patterns with multilobe peripheral and posterior lung distribution, or central peripheral distribution (Figure 7) [8,131,132,133,134]. In a recent meta-analysis, Garg et al. [131] reported that the pooled prevalence of GGO was 66.9%, consolidation was 32.1%, GGO plus consolidation was 44.9%, and crazy paving was 23.6%. Li et al. [132] found that the superior and middle lobes were more involved in severe cases. Other typical findings include small vessel enlargement, pleural thickness, bronchial distortion, and fibrosis (Figure 7) [36,128,133]. Less common findings include reverse halo sign and halo sign (Figure 8) [28,119,131,134]. Small vessel thickness is considered a typical finding in the early phase of the disease, and bronchiectasis and fibrosis in the late phase [128,134,135]. The reverse halo sign is reported by some authors in the later phase of the disease [28,136]. COVID-19 pneumonia shares a lot of CT features with OP, such as peripheral GGO, consolidation, or both in bilateral multifocal distribution, and reverse halo sign, and these features were also confirmed on histopathological examination [137,138,139]. To provide standardized communication, chest CT findings for COVID-19 pneumonia are currently classified as typical, indeterminate, and atypical [13,15,16,19,136]. The COVID-19 imaging reporting system (CO-RADS) was another method proposed to grade CT findings based on typical and atypical features and low to moderate or high levels of suspicion of COVID-19 [140]. Following a structured reporting system, including all potential findings of COVID-19 pneumonia, has been suggested by the major radiology societies (Table 2) [13,15,16,19,136].

### 4.4. COVID-19 Pneumonia CT Staging

Dynamic changes of the disease have been identified mainly based on a Chinese longitudinal study during the short follow-up period (Figure 9). Zhou et al. [135] identified three stages: early rapid progressive stage (1–7 days); advanced stage (8–14 days), characterized by the coexistence of signs of progression and absorption; and advanced stage (>14 days), during which increased signs of repair appear as subpleural lines, bronchial distortions, and fibrotic stripes. Pan et al. [108] reported four stages (0–4, 5–8, 9–13, and >14 days) based on the degree of lung involvement from day 0 to 26 after disease onset. Wang et al. [140] described six temporal phases based on the number of days from the onset of symptoms (<0, 0–5, 6–11, 12–17, 18–23, and >24 days) with a peak on days 6–11. Barenhaim et al. [28] reported the time between initial symptoms and subsequent chest changes on CT as early (0–2 days), intermediate (3–5 days), or late (6–12 days). In the early phase, GGO alone or GGO plus a reticular pattern or consolidation is a common finding. In this phase, early involvement of parenchyma around the bronchioles has been reported, followed by secondary pulmonary lobule with diffuse alveolar damage [133,135]. Small vessel enlargement can be correlated with focal lung vasculitis caused by inflammatory cytokines induced by the virus that increase vessel permeability [128,139,141,142,143]. This feature can also be correlated with the presence of microthrombi found on histopathological examination in post-mortem studies [139,143]. The progressive phase is usually characterized by extension of GGO or crazy paving and initial consolidation. In the peak or advanced stage, dense consolidation becomes more predominant. In the progressive phase, if the patient’s immunity is low or there is no response to therapy, COVID-19 pneumonia can progress to ARDS [133]. In the absorption phase, lung alterations start to decrease and signs of fibrosis with fibrotic streaks, traction bronchiectasis, bronchial distortion, and subpleural fibrotic line can be predominant [28,109,133,135,138].

A change in lung fibrotic alterations with residual GGO, consolidation, and interstitial thickness can be found in patients who were discharged after having COVID-19 [24,115,116]. However, few longitudinal studies have explored this issue. Fu et al. [117] demonstrated that residual lung alterations persisted even after two consecutive negative RT-PCR tests in patients with moderate and critical COVID-19, consisting of mixed GGO and consolidation, interstitial thickening, and pleural effusion. Pulmonary fibrosis was mainly seen in the critical group. Liu et al. [114] described three patterns of residual lung alterations: the first consisted of GGO that gradually disappeared; the second consisted of a mixed pattern in which fibrous stripes developed within the GGO area, followed by gradual resorption and disappearance; and in the third, there may have been some residual fibrous stripes that gradually reduced over time.

Urciuoli et al. [116] described six confirmed cases of patients with COVID-19 pneumonia who underwent follow-up CT four months after the onset of symptoms. In this case series, some patients presented respiratory complaints and only one presented complete resolution of lung alterations. Two cases presented a mixed pattern, one presented fibrotic stripes, one presented a mixed pattern with patchy GGO, and one presented a fibrotic pattern. In a study by Zhao et al. [115], residual lung abnormalities were found in 25.45% of COVID-19 patients after three months, and 30.9% of patients continued to show gastrointestinal symptoms. Yu et al. [144] found signs of fibrosis on CT in almost half of the discharged patients and reported that CT could also have predictive value for fibrotic development. In that study, COVID-19 patients with fibrosis took a longer time to recover than patients without fibrosis and had a higher rate of admission to the ICU and higher levels of inflammatory indicators during follow-up.

### 4.5. CT Severity Scores as Prognostic and Predictive Indicators of Clinical Outcome

The prognostic value of CT has been reported by several studies [145,146,147,148,149,150,151,152,153,154,155,156,157,158,159,160]. The use of a CT severity score (CT-SS) may be useful to standardize the assessment of lung alterations in COVID-19 pneumonia and to stratify patient risk and predict short-term outcomes [108,145,148,150,151,152]. The CT-SS was previously reported as a risk factor for mortality in ARDS [153]. Different visual versions of the CT-SS have been proposed with semi-quantitative methods (Table 3). Most of them were previously used in the evaluation of patients with severe acute respiratory syndrome (SARS) [134].

Pan et al. [109] found that CT-SS (Figure 10) was correlated with disease staging. Francone et al. [145], with the same CT-SS, found higher CT scores for critical COVID-19 patients, with a score ≥18 predictive of death.

Zhan et al. [154] described four patterns of COVID-19 pneumonia evolution on CT based on disease staging and CT-SS, with higher scores for patients with a more prolonged course of the disease. Other authors, applying the same CT-SS, found higher scores for critical COVID-19 patients and patients who were admitted to the ICU [132,153]. Khosravi et al. [150] showed that baseline CT-SS can predict adverse outcomes including days of recovery, ICU admission, and mortality; thus, it appeared to be poorly correlated with initial disease severity.

Yang et al. [155] found higher CT-SS for severe forms of COVID-19, with a positive predictive value (PPV) of 75% and a negative predictive value (NPV) of 96.3%. A study by Guillo et al. [108] found that patients with more than 25% lung alteration were intubated or deceased after three weeks. Wang et al. [149] suggested that CT-SS in combination with several clinical variables on admission can predict complications in hospitalized COVID-19 patients. A study by Li et al. [151] found that the CT-SS in the second week appeared to have predictive value for clinical severity. Therefore, these authors suggested including the CT score as one of the criteria to assess clinical severity together with clinical indicators. Liu et al. [156] also found that patients with severe and critical disease showed the greatest severity on the second follow-up CT, with a mean interval of 8 ± 3 days after the initial CT scan, and also showed a higher occurrence of the crazy paving pattern. A study by Hu et al. [157] showed that COVID-19 patients who died had an increased proportion of consolidations and higher CT-SS during the follow-up compared to the initial scans. Tabatabaei et al. [148] reported that CT-SS is a reliable predictive factor of mortality in previously healthy non-elderly individuals with COVID-19 pneumonia. Abbasi et al. [158] found a significant correlation between time of admission to time of death and CT-SS, and between CT-SS and time to ICU admission. These authors also found that mortality was significantly higher in patients with higher CT-SS, even after adjustments for clinical, demographic, and laboratory parameters. However, Feng et al. [147] recently found that CT-SS was associated with higher inflammatory levels, a higher neutrophil-to-lymphocyte ratio (NLR), and older age and negatively associated with lymphocyte counts on day 3 after admission, indicating a potential role in early prediction of lymphopenia. CT-SS on admission was also found to be an independent predictor of progression to severe COVID-19 pneumonia [147]. Li et al. [159] also suggested that decreased lymphocytes in severe/critical patients can reflect the consumption of a large number of immune cells.

Cao et al. [146] described that a higher CT score, older age, and lymphopenia were independent predictors of mortality risk. Pleural effusion was also found to be more common in patients with severe disease [146]. Therefore, the use of the visual CT-SS in clinical practice can allow rapid identification of COVID-19 patients at higher risk of developing ARDS (Figure 11), due to the discrepancy between normal saturation and the extensive lung alterations usually found on CT, in order to conduct earlier risk stratification management and provide earlier treatment [151]. However, several authors have recently proposed quantitative methods using open-source platforms to evaluate CT-SS and found that compared with the semi-quantitative visual score, the quantitative CT parameters have superior accuracy (Figure 12) [152,159]. Deep learning algorithms are also promising in the evaluation and quantification of COVID-19 pneumonia [160,161].

## 5. Imaging and Histopathological Correlations

Few studies have correlated imaging findings in severe cases of COVID-19 pneumonia with histopathological findings [162,163,164,165,166,167]. This issue was investigated especially in case reports and case series mainly based on CT features described in ante-mortem or post-mortem examinations [163,164,165,166]. CT imaging findings such as GGO and consolidations usually correspond to histopathological examinations with diffuse alveolar damage (DAD) in different stages [162,163,164,165,166]. DAD is usually a nonspecific response of the lung to a multitude of injurious agents characterized by edema, endothelial and alveolar injury with hyaline membrane formation in the acute phase, and fibroblast proliferation and interstitial fibrosis in the organizing phase [162,163].

Henkle et al. [162] correlated the histopathological findings of 14 patients who died from COVID-19 confirmed by RT-PCR with ante-mortem CT examination. They found that the presence of GGO correlated with capillary dilation and congestion, interstitial edema, hemorrhage, and acute DAD. The enlarged pulmonary arteries and CT vascular signs such as small vessel enlargement were explained by severe microangiopathy, confirming the importance of vascular alterations. The acute DAD had an approximate duration of 7 days. Bronchial wall thickening and consolidations on CT corresponded to superimposed acute bronchopneumonia. These features increased in the proliferative phase of DAD, which was also characterized by fibroblast proliferation within the interstitium and alveoli. No histopathological features of organized pneumonia were found in that study. The case series of Recald-Zamacona et al. [163] confirmed these correlations.

Post-mortem CT may be useful to identify any cause of death in patients with suspected COVID-19. Suess et al. [165] described the CT features of a 59-year-old patient with COVID-19 diagnosed by RT-PCR that initially showed good clinical conditions; however, in a few days, he had rapid deterioration of respiratory symptoms and died a few days later at home. The post-mortem CT revealed bilateral GGO with consolidations and gross section specimens of the lung showed pulmonary edema with hemorrhage on the pleural surface without signs of pleurisy. Immunohistochemical staining showed severe type II pneumocyte hyperplasia with large nuclei and viral cytopathic-like changes. Increased numbers of vascular megakaryocytes and interstitial lymphocytic cells were also described. Furthermore, in this case, the authors did not find any evidence of organizing pneumonia.

Ducloyer et al. [165] described the case of a 75-year-old man who presented to a French hospital with 4 days of fever whose oropharyngeal and nasopharyngeal (OP/NP) swabs were positive for COVID-19. Based on its clinical stability, he returned home; however, a few days later, he was found deceased in his bed. On post-mortem examination, bilateral pneumonia with a crazy paving pattern and an extension of 85% of the lung parenchyma was found. On histopathological examination, diffuse alveolar damage in different stages was found, including an acute stage characterized by diffuse hyaline membranes associated with alveolar edema and an organized stage with enlargement of alveolar septa, alveolar fibrin deposits, and hyperplasia of type 2 pneumocytes. Almeida Mointer et al. [167] correlated post-mortem CT images with histological findings in fatal cases of COVID-19. Most of the histopathological findings described in fatal cases of COVID-19 pneumonia are similar to those found in severe acute respiratory syndrome corona virus-1 (SARS-CoV-1), Middle East respiratory syndrome coronavirus (MERS-CoV), and also Ebola virus [168].

## 6. COVID-19 Complications

Pulmonary manifestations in COVID-19 can be viewed as a combination of viral pneumonia and ARDS [167]. COVID-19 pneumonia can progress into ARDS in 42% of cases, and ARDS is one of the main causes of death in COVID-19 patients, especially in the ICU [169,170]. Higher levels of inflammatory cytokines are usually found in COVID-19 patients with more severe disease, and vascular enlargement is typically seen in patients with COVID-19 [139,143,170,171,172]. ARDS is a heterogeneous clinical syndrome that can be caused by various factors. COVID-19 ARDS (CARDS) is diagnosed when someone with confirmed COVID-19 infection meets the Berlin 2012 ARDS diagnostic criteria that consist in the presence of these features: acute hypoxemic respiratory failure, presentation within 1 week of worsening respiratory symptoms, bilateral airspace disease on CXR or CT or LUS that is not fully explained by effusions, lobar or lung collapse, or nodules, and exclusion of cardiac failure. To date, the exact mechanism of CARDS remains unclear, and it has been shown to have different findings from typical ARDS [169,173,174,175,176]. Various studies have suggested that patients with CARDS have markedly higher lung compliance than patients with typical ARDS, and CARDS has frequently been associated with pulmonary thrombotic injury [143,169,173,174,175,176,177,178]. Chiumiello et al. [177] showed that patients with CARDS exhibit normal pulmonary compliance with severe hypoxemia, probably sustained by a mismatch between hypoxic pulmonary vasoconstriction (HPV) and ventilation/perfusion (V/Q).

Various researchers are focusing on the vascular-centric pathogenesis of COVID-19 and whether the atypical features of CARDS can be due to vascular damage [143,172,173,174,175,176,177,178,179,180]. These findings were also confirmed in post-mortem studies [139,143]. In particular, Borczuk et al. [143], in a multicenter study on 65 patients who died due to COVID-19, reported that vascular injury of pulmonary vessels was a common finding as well as systemic vascular alterations such as thrombotic microangiopathy in other organs, including the kidney, liver, heart, and central nervous system. Furthermore, disseminated intravascular coagulopathy (DIC) has been reported as another complication and cause of death in COVID-19 infection [139,143,179,180]. Patients with CARDS share similar findings with those observed in sepsis, such as vasoplegia or persistent hypotension, suggesting that the primary insult is to the pulmonary endothelium [172,173]. Therefore, CARDS may be related to cytokine storms, coagulation dysfunction, and microvascular thrombosis. Nevertheless, a toxin-like mechanism should be explored. On the other hand, pulmonary thromboembolic complication is a common finding in COVID-19 and was found in around 23–30% of cases, and the incidence of venous thromboembolic disease (VTE) was found in 25% of cases; these features are more common in ICUs [181,182,183]. All of these vascular pulmonary alterations found in COVID-19 patients can lead to pulmonary hypertension, which is associated with an increased risk of death [182].

Other common chest complications of COVID-19 pneumonia include PNX, PM, pneumopericardium, and subcutaneous emphysema, which are usually secondary to pulmonary barotrauma from mechanical ventilation in the ICU (Figure 13) [184]. They are usually associated with a long hospital stay and higher mortality and occurred in 15 to 40% of COVID-19 patients who underwent mechanical ventilation [185,186]. Mechanical ventilation with high positive end-expiratory pressure (PEEP) can increase high intra-alveolar pressure, causing alveolar rupture through the disproportionate distribution of volume and pressure from the ventilator with air dissection along the bronchovascular sheets toward the mediastinum, leading to PM. Occasionally, air in the mediastinum can escape into the pleural space and cause pneumothorax (PNX), or through the parietal pericardium and cause pneumopericardium. Air can also travel toward the thoracic inlet and into neck soft tissue, causing subcutaneous cervical-facial emphysema, and into the peritoneal spaces of the abdomen, causing pneumo-retroperitoneum [185,186,187]. Gattinoni et al. [178] found that in critically ill COVID-19 patients, the incidence of PNX increased in those who were mechanically ventilated for a long period of time.

Spontaneous pneumomediastinum (SPM) and spontaneous pneumothorax (SPX) represent rare complications of COVID-19 and were mainly reported in case reports and case series [187,188,189,190,191,192]. SPM is usually a benign entity and a possible complication of severe COVID-19 pneumonia. SPM may be linked to the Macklin effect, which consists of the progression of air originating from an alveolar rupture, causing interstitial emphysema, which may further progress to the mediastinal space and cause subcutaneous emphysema [188]. It has been reported that some patients who developed SPM had been under treatment with continuous positive airway pressure (CPAP) or high-flow oxygen therapy [186,187,191,192,193]. It has also been reported that corticosteroid use may contribute to the development of SMP by weakening the pulmonary interstitial tissue [184,186]. SPX without SPM can be due to structural modification, such as cystic and fibrotic changes, that can occur in COVID-19 pneumonia [185,186,187]. Zantah et al. [186] reported six cases of SPX in 902 patients with COVID-19 pneumonia with an incidence of 0.66%. Lung cavitations have also been reported as rare complications in COVID-19 pneumonia and can be due to cavitation of pulmonary infarctions or caused by destructive pulmonary fibrosis in the late phase of the disease, or associated with necrotizing pneumonia [194,195].

Imaging plays a pivotal role in the evaluation and management of COVID-19 complications. CXR is the first-line imaging modality to evaluate tube positions in the ICU and associated complications. Chest CT is the best imaging tool to understand the pathophysiology of ARDS and to visualize pulmonary vascular alterations, as it well depicts signs of pulmonary hypertension and can rule out lung thromboembolism with angiography acquisitions [127,128,171]. Chest CT is also more accurate than CRX in evaluating barotrauma, as it can often detect occult complications and allow visualization of the interstitial PNX that leads to SPM through the Macklin effect [188]. In addition, LUS can also be a practical approach to detect ARDS in the ICU. A brief summary of all the different imaging modalities used for COVID-19 pneumonia diagnosis, including clinical complications, has been reported in Table 4.

## 7. Conclusions

In conclusion, in this review, we tried to summarize in a practical way the current findings and indications of imaging modalities including CXR, LUS, and CT in the detection and management of COVID-19 pneumonia, and the scores proposed for each modality to assess the severity of extension of COVID-19 pneumonia and its complications. Chest imaging plays an important role in the pandemic context. However, the choice of what imaging modality should be used, either CXR or CT, is usually left to the judgment of the clinical team, the availability of local resources, and the expertise of radiologists. Emerging studies suggest a potential role for CT as a prognostic and predictive indicator of COVID-19 patient outcomes. COVID-19 pneumonia can lead to many lung complications, such as PMS, PMX, cavitations, and atypical ARDS. The vascular damage with cytokine storm activation can lead to the atypical features of CARDS. Most of the histopathological findings described in COVID-19 are similar to those found in severe acute respiratory syndrome corona virus-1 (SARS-CoV-1), Middle East respiratory syndrome coronavirus (MERS-CoV), and also Ebola virus. However, a toxin-like mechanism should also be explored.

## Figures and Tables

**Figure 1 diagnostics-11-00437-f001:**
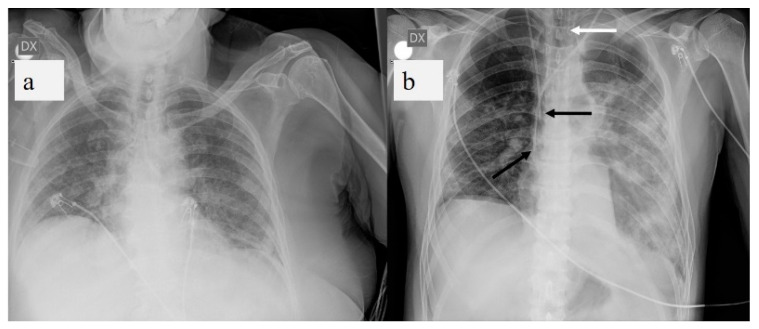
(**a**) Chest X-ray (CXR) of COVID-19 patient at bedside on anterior-posterior projection shows diffuse linear opacities associated with ground-glass opacity (GGO). (**b**) CXR of patient admitted to intensive care unit (ICU) showing two central venous catheters in superior vena cava (black arrows) and endotracheal tube (white arrow). Diffuse confluent GGO with consolidation in left lung and linear opacity and GGO in right lung are also visible.

**Figure 2 diagnostics-11-00437-f002:**
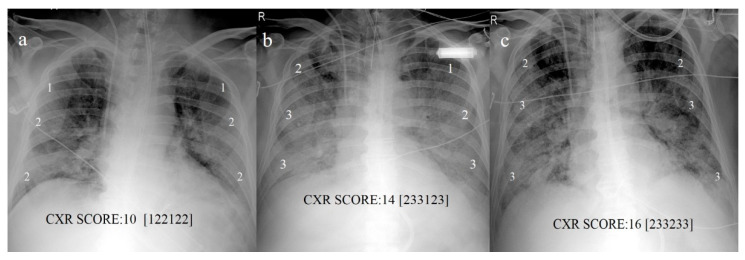
CXR scores based on scores proposed by Maroldi et al. [27] in COVID-19 pneumonia. (**a**) Baseline score of 10 on CXR at bedside in AP projection at admission; (**b**) increased score of 14 reveals progression of COVID pneumonia on CXR follow-up on day 13; (**c**) score of 16 indicates further progression on day 18.

**Figure 3 diagnostics-11-00437-f003:**
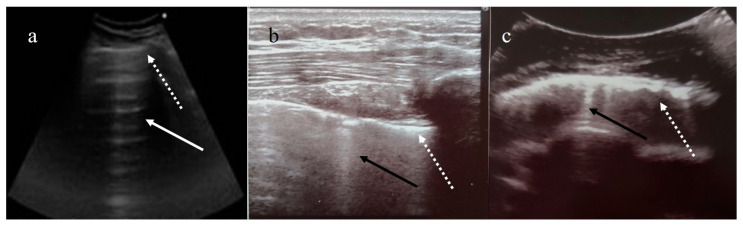
(**a**) Horizontal A line artifacts (solid white arrow) as regular aeration in a healthy lung, and normal pleural line (dashed white arrow); (**b**) B line vertical artifact (black arrow) in healthy lung with linear probe, and normal pleural line (dashed white arrow); (**c**) multiple B line artifacts (black arrow) with diffuse pleural thickness (dashed white arrow) in a patient with fibrosis. (Images courtesy of Dr. Luigi Monaco, head of the ultrasound unit at San Giuseppe Moscati Hospital.)

**Figure 4 diagnostics-11-00437-f004:**
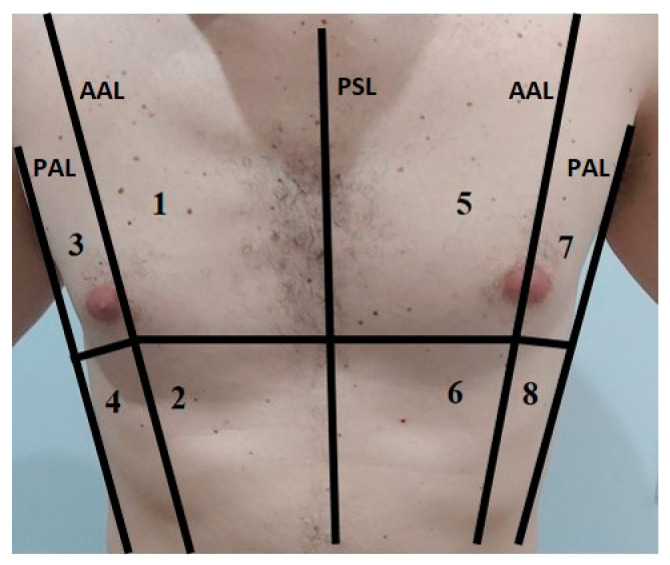
In this image is represented an example of the anatomical subdivision of the chest by 3 lines (median: parasternal line PSL; lateral: anterior axillary lines AAL; posterior: posterior axillary lines PAL) into 8 zones for the LUS examination.

**Figure 5 diagnostics-11-00437-f005:**
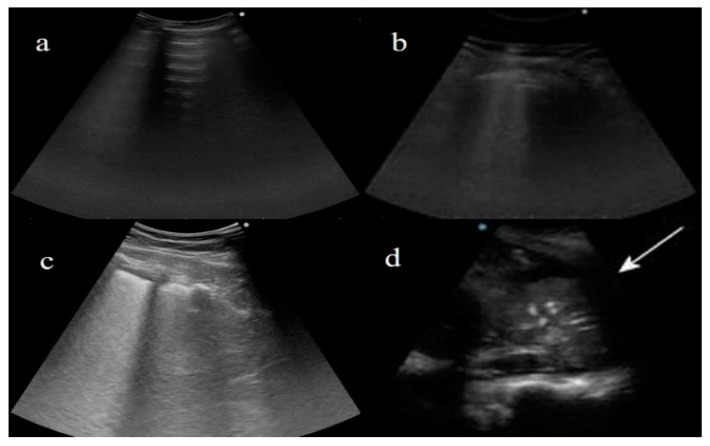
Examples of lung ultrasound score (LUSS) based on score proposed by Dargent et al. [61] for patients with COVID-19 pneumonia. (**a**) LUS describes horizontal A lines for regular aeration (score: 0); (**b**) multiple B lines arising from thickened pleural line for moderate loss of aeration (1—B1); (**c**) coalescent B lines or white lung for severe loss of aeration (2—B2); (**d**) consolidation, with air bronchogram, for absence of aeration (3) and pleural effusion (white arrow).

**Figure 6 diagnostics-11-00437-f006:**
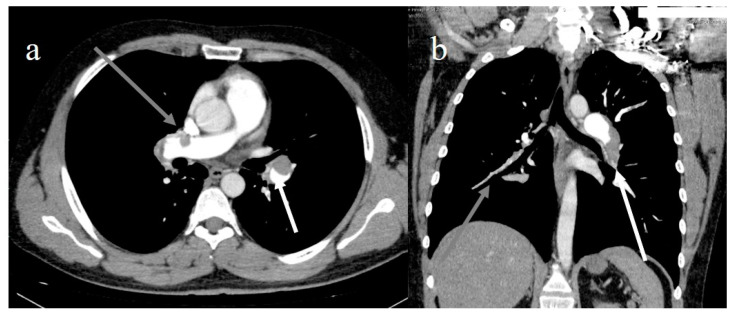
(**a**) Lung embolisms in left pulmonary artery (white arrow) and right pulmonary artery (gray arrow); (**b**) extensive lung pulmonary embolism in left pulmonary artery, in lobar branch of inferior lobes (white arrow), and in lobar and segmentary branches of right inferior lobe (gray arrow).

**Figure 7 diagnostics-11-00437-f007:**
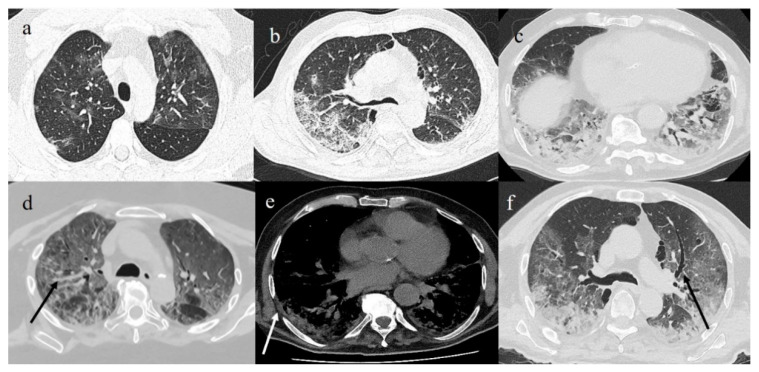
(**a**) GGO pattern in COVID-19 pneumonia with peripheral and central distribution; (**b**) crazy paving pattern on right side with peripheral and posterior distribution; (**c**) consolidation areas in inferior lobes; (**d**) diffuse vascular enlargement (black arrow); (**e**) pleural thickness (white arrow); and (**f**) bronchiectasis (black arrow).

**Figure 8 diagnostics-11-00437-f008:**
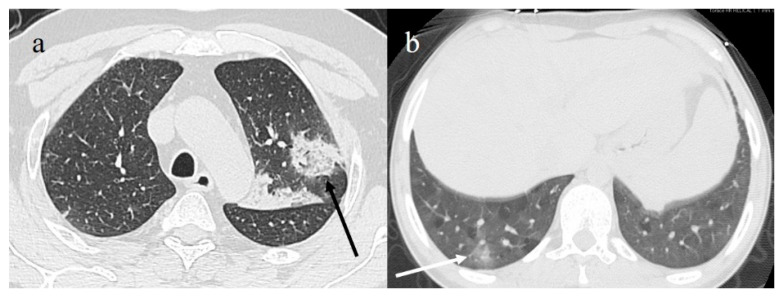
(**a**) Reverse halo sign (black arrow) consisting of central ground-glass opacity surrounded by complete ring of consolidation; (**b**) GGO areas surrounded by small consolidative area as halo sign (white arrow).

**Figure 9 diagnostics-11-00437-f009:**
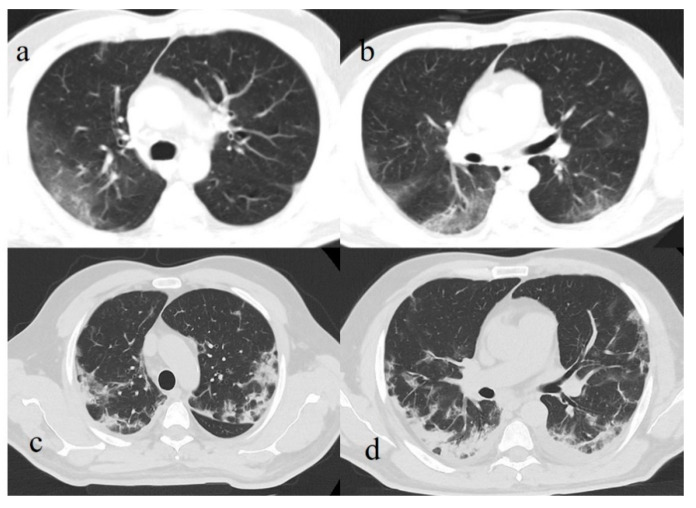
(**a**) Initial GGO pattern of COVID-19 pneumonia involving right superior lobe with peripheral distribution; (**b**) some GGO areas with interstitial thickness in both inferior lobes; (**c**,**d**) same areas in (**a**,**b**) evolving in consolidations.

**Figure 10 diagnostics-11-00437-f010:**
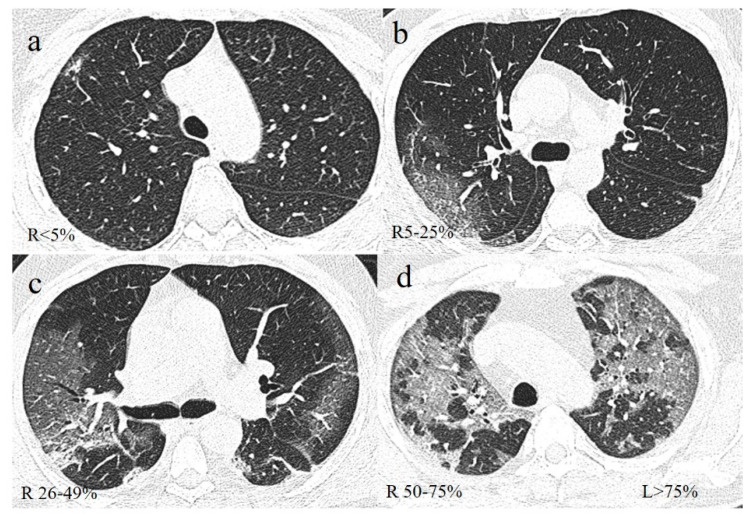
Example of lung involvement in COVID-19 pneumonia based on CT-SS proposed by Pan et al. [108]. (**a**) GGO pattern in right (R) superior lobe with parenchymal involvement of <5%; (**b**) GGO in right superior lobe with parenchymal involvement of 5–25%; (**c**) GGO in right superior lobe with parenchymal involvement of 26–49%; (**d**) GGO with parenchymal involvement of 50–75% in right superior lobe and >75% in left (L) superior lobe.

**Figure 11 diagnostics-11-00437-f011:**
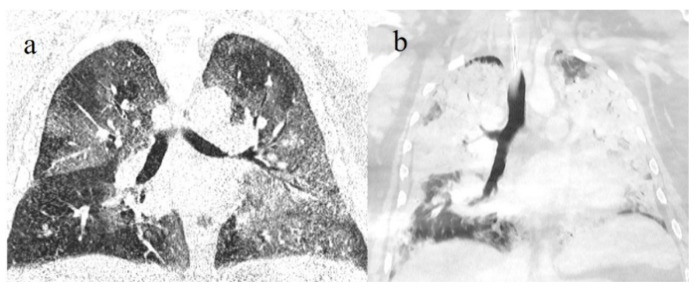
(**a**) COVID-19 pneumonia on initial examination of an obese young patient with severe lung involvement (CT-SS of 18) that (**b**) further progressed to acute respiratory distress syndrome (ARDS) with diffuse consolidations.

**Figure 12 diagnostics-11-00437-f012:**
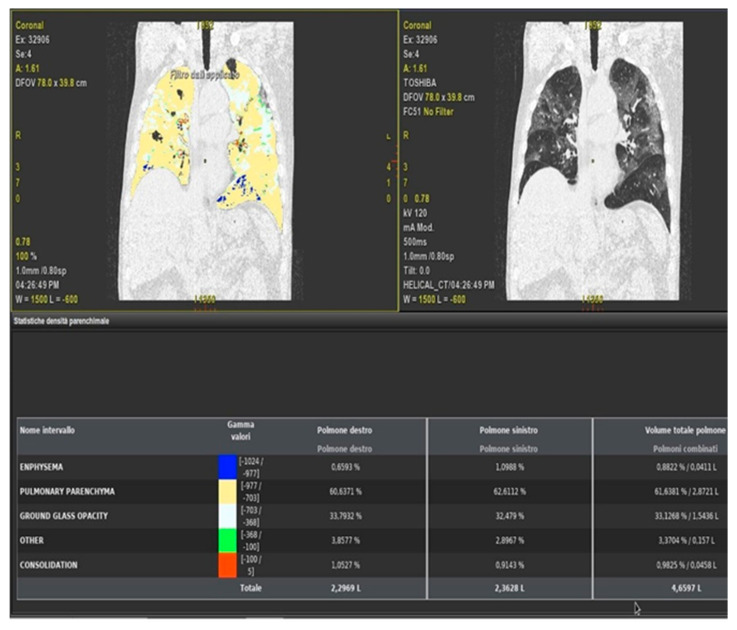
Example of quantitative method using Thoracic-VCAR software that evaluates percentages of GGO, consolidations, and pulmonary parenchyma without COVID-19 pneumonia involvement.

**Figure 13 diagnostics-11-00437-f013:**
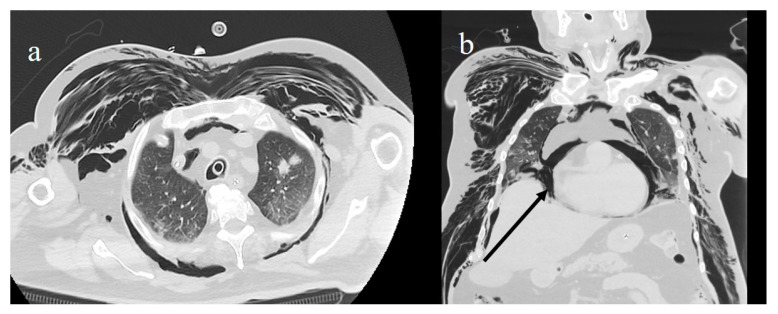
(**a**) Diffuse subcutaneous emphysema and pneumomediastinum in a COVID-19 patient who underwent mechanical ventilation; (**b**) pneumopericardium (black arrow).

**Table 1 diagnostics-11-00437-t001:** Main indications of international imaging guidelines for COVID-19: chest X-ray (CXR), lung ultrasound (LUS), and chest computed tomography (CT).

Radiology Societies with Consensus Statements on Imaging Guidelines for COVID-19	CXR	LUS	CT
**American College of Radiology**	Portable radiography units when CXR is considered medically necessary	No data	Only in symptomatic and hospitalized patients with specific clinical indications
**British Society of Thoracic Imaging**	Clinically stable patients with fever and respiratory symptoms if clinically required; for critically ill patients	Monitor critically ill patients	Seriously ill patients with uncertain or normal chest X-ray findings and if any complication is suspected during follow-up; if RT-PCR is not available
**Canadian Association of Thoracic Radiology/Canadian Association of Radiologists**	CXR may be useful in patients presenting with moderate to severe symptoms; in emergency department when RT-PCR assay is not available	No data	Low-dose CT only if results are expected to influence patient management or in high-risk individuals; CT pulmonary angiography in setting of suspected acute pulmonary embolism
**Chinese Society of Radiologists**	Follow-up for critically ill patients; lower sensitivity than CT for evaluation of early stage of pneumonia	Emergency and critical care setting	Chest CT is the most valuable imaging tool for clinical diagnosis of early-stage COVID-19 pneumonia when symptoms are nonspecific; chest CT can also evaluate time course and assess evolution of disease severity
**European Society of Radiology and European Society of Thoracic Imaging**	For ICU and in patients that are too fragile to be sent to CT	At bedside for pregnant women, children, ICU patients	In patients with respiratory symptoms such as dyspnea and desaturation; allows evaluation of disease extent at baseline, which may help predict poor outcome and need for ventilation
**Fleischner Society**	The choice of imaging modality is left to the judgement of clinical teams. CXR is usually preferred as the first imaging tool; however, it has lower sensitivity than CT. CXR is indicated in a resource-constrained environment where access to CT is limited. Daily chest radiographs are NOT indicated in stable intubated patients with COVID-19	Not suggested for limited experience	Choice of imaging modality left to judgment of clinical team; CT usually indicated for patients with functional impairment and/or hypoxemia after recovery or for evaluation of complications
**Italian Society of Medical and Interventional Radiology**	First overview of the patients, especially in the emergency room; in hospitalized patients and in ICU	Critically ill patients	CT may be useful for monitoring lung involvement and managing suspected cases
**Royal Australian and New Zealand College of Radiologists**	In hospitalized patients	No data	In patients with chronic or acute disease
**Royal College of Radiology**	Critically ill patient	No data	In seriously ill patients

**Table 2 diagnostics-11-00437-t002:** Example of structured report for COVID-19 pneumonia based on European Society of Radiology ESR/European Society of Thoracic Imaging (ESTI) and Italian Society of Medical and Interventional Radiology (SIRM).

Structured Report Example
**Technique**: The examination was performed with unenhanced volumetric low-dose high-resolution (HRCT) technique with DLP: (mGy.cm) **Indication**: COVID-19 suspicion/initial assessment/follow-up **Findings**: Report parenchymal findings: **Presence of typical/atypical parenchymal findings**: **Typical findings**: Presence of GGO, crazy paving, consolidation with distributions in the axial plane (peripheral or peripheral with central distributions). Other findings such as air bronchogram, bronchial distortion or bronchiectasis, vascular enlargement, micro- or macronodules, pleural thickness, pleural effusion. Visual assessment for extension may be used [13], or quantification of GGO and consolidation area with software [16,47]. **Atypical findings**: Presence of tree-in-bud pattern/centrilobular nodules/endobronchial secretion/lobar or segmental consolidation. Any co-existing lung pathologies (lung emphysema, fibrosis) or complications (such as presence of barotrauma or SPM or SPNX). Any mediastinum findings: presence of adenopathy, pericardial effusion, and pulmonary trunk diameter. **Conclusion**: CT findings typical/indeterminate/atypical of COVID-19 (CO-RADS levels ranging from very low risk (CO-RADS 1) to very high suspicion (CO-RADS 5), with category 6 reserved for proven RT-PCR cases), with limited, moderate, or severe disease extent.

**Table 3 diagnostics-11-00437-t003:** Main semi-quantitative methods used to assess CT severity score (CT-SS) (second column) and correlations between CT findings or outcomes in patients with higher CT-SS (third column) found in published articles (first column, which also specifies number of patients in each study).

Paper	Number of Patients	CT Severity Scores (CT-SS) in COVID-19 Pneumonia	Correlations of Higher CT-SS and CT Findings/Outcomes
Abbasi et al. [158]	262 patients	Degree of involvement in each zone scored as follows: - 0: no involvement - 1: <25% involvement - 2: 25 to <50% involvement - 3: 50 to <75% involvement - 4: ≥75% involvement (CT-SS 0–24)	CT-SS can discriminate admitted patients with higher risk of in-hospital mortality with acceptable accuracy (area under the curve, 0.839). Mortality was significantly higher in patients with higher CT severity score even after adjustment for clinical, demographic, and laboratory parameters.
Khosravi et al. [150]	121 patients		Patients with baseline CT-SS > 8 had 3-fold higher risk of poor outcome (ICU admission, intubation, mortality).
Li et al. [151]	53 patients		Higher CT-SS in severe/critical patients with higher GGO in second week, higher consolidation and crazy paving score in third week. Overall lung involvement score in second week appeared to have predictive value for whole-course clinical severity with optimal cut-off of 5.25 points.
Chung et al. [154]	21 patients	Each lung lobe scored using 0–4 Likert scale: - 0: no involvement - 1: 1–25%, minimal involvement - 2: 26–50%, moderate involvement - 3: 51–75%, moderate–severe involvement - 4: 76–100%, severe involvement (CT-SS 0–20)	Higher CT-SS for patients in ICU.
Hu et al. [157]	73 patients		Moderate positive correlation between CT severity scores and inflammation-related factors of leucocytes, neutrophils, and IL-2R. CT-SS of lung involvement for patients who died from COVID-19 was significantly greater compared patients with mild to moderate disease.
Li et al. [132]	78 patients		Higher CT-SS (range of 8–18) in the severe critical type compared with the common type (range 1–11).
Liu et al. [156]	53 patients		In severe and critical group, GGO, fibrosis, and pleural thickening or adhesion could be found in every follow-up CT and were main signs in the two CTs. Right lung more involved in severe and critical group.
Tabetabei et al. [148]	30 patients		CT-SS ≥7.5 has highest sensitivity and specificity in ROC curve to predict mortality.
Zhan et al. [153]	110 patients		Higher CT-SS for patients with more prolonged disease course.
Francone et al. [145]	130 patients	Each lung lobe scored on a scale of 0 to 5: - 0: no involvement - 1: <5%, minimal involvement - 2: 5–25%, mild involvement - 3: 26–49%, moderate involvement - 4: 50–75%, moderate–severe involvement - 5: >75%, severe involvement (CT-SS 0–25)	Death of patients with CT-SS ≥ 18.
Li et al. [159]	83 patients		Severe/critical patients were older and had more underlying diseases than others. Decreased lymphocyte count in severe/critical patients.
Pan et al. [109]	21 patients		CT-SS correlated with disease stage.
Guillo et al. [108]	214 patients	Severity of COVID-19 pneumonia graded as minimal (<10% lung parenchyma), moderate (10–25%), intermediate (25–50%), severe (50–75%), critical (50–75%).	68 % of patients with disease extent exceeding 25 % of the lung parenchyma were intubated or deceased in the 3 weeks following CT.
Yang et al. [155]	102 patients	Considered 20 lung regions, assigning scores for parenchymal opacification of 0 (0% involvement of each region), 1 (<50% involvement), or 2 (>50% involvement) (CT-SS 0–40).	Higher CT-SS in patients with severe COVID-19 disease with CT-SS of 19.5 for identifying severe cases with a PPV of 75% and an NVP of 96.3%.
Wang et al. [149]	161 patients	CT visual severity levels: - None or mild: <50% involvement - Moderate: 50–75% involvement - Severe: >75% involvement	Higher CT-SS were associated to the severity clinical course. Non-survivors showed much higher CT-SS compared with survivors, without a visually apparent decrease between week 1 and week 2.

**Table 4 diagnostics-11-00437-t004:** General indications based on current literature for chest X-ray (CXR), chest CT (CT), and lung ultrasound (LUS) for detection and management of COVID-19 pneumonia, with advantages and limitations. ARDS, acute respiratory distress syndrome; CT, computed tomography; FU, follow-up; ICU, intensive care unit; LUS, lung ultrasound; PMS, pneumomediastinum; SPM, spontaneous pneumomediastinum; PX, pneumothorax; SPX, spontaneous pneumothorax; TE, thromboembolism.

COVID-19 Imaging Tools	General Indications	Advantages	Limitations
**CXR**	For symptomatic stable patients in ED; for patients in ED at moderate–high risk of progression, choice between CXR and CT based on judgment of clinical team, availability of local resources, and expertise of radiologists; in the ICU to evaluate complications (PMS, PX, ARDS) and chest tube positioning	Low cost, portable, lower dose burden than CT	Lower sensitivity than CT for evaluating COVID-19 pneumonia, especially in early phase; inadequate information on specificity
**Chest CT**	In ED in presence of high pretest probability for symptomatic patients with comorbidities or functional impairment and during FU for patients at moderate–high risk of progression; evaluation of fibrotic changes complications (barotrauma, SPM, SPX, ARDS, TE); CT can be indicated for symptomatic patients with multiple negative RT-PCR results; long-term FU	Easily available, rapid, high sensitivity in early phase of COVID-19 pneumonia, prognostic and predictive value in mortality through evaluation of pneumonia extension with CT-SS index; possible to visualize Macklin effect on CT; post-mortem evaluation	Low specificity, high dose burden, not used for screening asymptomatic patients or those with mild symptoms
**LUS**	For monitoring critically ill patients, especially in ICU	Low cost, portable, rapid, no radiation dose	Presence of air, low specificity, operator-dependent with intra- and inter-operator variability in B lines counted based on type and frequency of probe used and ultrasound machine setting

## Data Availability

Data sharing not applicable.

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
