# Peer review of "A Pictorial Review of the Role of Imaging in the Detection, Management, Histopathological Correlations, and Complications of COVID-19 Pneumonia"

_diagnostics, 2021, doi:10.3390/diagnostics11030437_

Round 1
Reviewer 1 Report
Very good job, congratulations.
Author Response
Dear Reviewer,
Thank you for your valuable comments and appreciations. We are grateful to you. We have added some corrections in the text that you can find in red on the Suggestions of two Reviewers. Therefore, we have added the table 1 that summarized the main International guidelines and the previous table 1 becomes the table 3 that summarized the general guidelines on the base of the current literature. We have highlighted also the limits of the LUS, we have added the Figure 3 and 4 and we have added the section 6 on the histopathological correlations. So, the new references were marked in red and the rearranged reference in red. All the modification marked in redWe hope that you can also find all the modifications satisfactoryWith Kind regards
Reviewer 2 Report
The pictorial review is interesting but incomplete. In particular, it does not emphasize the significant limitations of lung ultrasound . Most of the ultrasound studies cited were published as preprints, which do not undergo the same rigorous checks as published studies. The citations are incomplete. The ultrasound images are insufficient ( fig.3) and do not highlight the low specificity of the ultrasound patterns. The ultrasound citations are incomplete.
Lung ultrasound (LUS) is a real-time, low-cost, bedside, imaging tool. However, if it's true that in COVID-19 lesions are more likely to have peripheral distribution and bilateral involvement and are in most part localized in the lower lung, LUS examination is limited by the presence of air, that is "the worst enemy of ultrasound", preventing the visibility of also big lesions due to the interposition of also a very thin layer ( micron or mm) of air between the pleural surface and the lesion in the more superficial lung. Furthermore, due to the anatomic constraints of the thoracic cage, LUS can at best, with patient in a sitting position, explore only the 70% of the pleural surface: the retro-scapular area, the costo-vertebral junction region and the pleuropulmonary area of the mediastinum are not explorable with ultrasound. As a result, compared to a Chest-CT scan, that allows to evaluate the actual lung volume involved by the disease, LUS examination implies a concrete risk to underestimate the actual disease's extent and to miss the detection of some lesions.
Several recent case reports in literature claimed that the most important ultrasound sign in the early stage of SARS-CoV2 infection would be focal or coalescent B-lines, while an "alveolar interstitial syndrome" (i.e. a pattern of multifocal and confluent B-lines) is considered to be the main feature in the progressive stage and in critically ill patients. This appears to suggest that B-lines have gained widespread scientific acceptance as a marker of "interstitial edema". However, to our knowledge, no approved international recommendation/guideline reports this indication. That's why "B lines",are only artifacts generated by the physical interaction between the ultrasound beam and the different structures crossed by it. In particular, they are due to the presence of microbubbles of air/gas, mixed with liquid film/edema and/or fibrosis, which resonate when crossed by the ultrasound beam. As a confirm, B-lines can be seen posterior to areas where fluid and gases are collected, for example in the bowel loops. Interfaces of this type are likely to be more numerous in a number of pleuro-pulmonary diseases, such as those associated with increases in the fluid component of the lung or in the proportion of tissue that is fibrotic: heart failure, ARDS, hydropneumothorax, fibrosis, emphysema, exacerbations of chronic obstructive pulmonary diseases, nephrotic syndrome, pleural effusion also minimal and lymphagitis. Furthermore, B-line can be seen even in healthy lungs, typically in the dependent regions (where the hydrostatic pressure gives a more fluid-rich interstitium), and in the residual cavity of patients who had undergone pneumonectomy. In fact, the postpneumonectomy space contains gas (air), fluid (liquid effusion) and/or fibrotic tissue, which are the physical basis for the creation of such artifacts. In addition, we should also consider that this pattern, suggestive for COVID-19, can instead hide another viral pneumonia, such as influenza A. In this pandemic period of virus spread, an ultrasound diagnosis of COVID-19 based on these signs is more likely to be "statistically" correct, but when the incidence of this pneumonia will become stable in the population, the risk of false positives and consequent misdiagnosis will increase. Indeed, the Fleischner Society and the Centers for Disease Control (CDC) recommend confirmation with viral tests, even in case of radiologic findings suggestive for COVID-19 at chest-CT
An artifactual LUS pattern, consisting in a B-lines' perceptual increase and pleural line abnormalities, is potentially misleading, not only because it may be found also in several types of underlying pleura-pulmonary conditions but also because it may be variable according to different intra- and inter-operator B-lines perceptual counting and ultrasound scan settings. The simple change of positioning of the probe with respect to the curvature of the patient's chest can modify the perceptual semi-quantitative evaluation of B-lines, that may, therefore, be very variable in different intra- and inter-operator examination. Also the increase in the pleural line movement rate can modify the perception of B lines! Furthermore, in this manuscript is not specified the type and frequency of the probe used, degree of total gain compensation (TGC), electronic focus and tissue harmonics set in the cited studies. These are all criteria causing bias in the US examination.
A CT scan may be the indicated modality for particular patient groups (e.g. those with suspected thrombotic/thromboembolic disease, multisystemic disease and to detect comorbidities, particularly in severe COVD-19 patient groups). As growing body of literature is confirming the multi-systemic nature of COVID-19 (including the nervous, vascular and cardiac systems, kidneys) , this raises questions on whether/when/how imaging other than that of the chest (e.g., cardiac ultrasound, brain MRI, vascular imaging, abdominal imaging) may contribute to early diagnosis and/or management of patients with COVID-19.
Future diagnostic accuracy studies should predefine positive imaging findings, include direct comparisons of the various modalities of interest on the same participant population, and implement improved reporting practices.
References
Radiological Society of North America Expert Consensus Statement on Reporting Chest CT Findings Related to COVID-19. Endorsed by the Society of Thoracic Radiology, the American College of Radiology, and RSNA. Radiology: Cardiothoracic Imaging 2020;2(2). doi: 10.1148/ryct.2020200152
Radiological Society of North America Expert Consensus Statement on Reporting Chest CT Findings Related to COVID-19. Endorsed by the Society of Thoracic Radiology, the American College of Radiology, and RSNA J Thorac Imaging. 2020 Jul;35(4):219-227.
Diagnosis of Coronavirus Disease (COVID-19) Pneumonia: Is Lung Ultrasound the Better Choice? AJR Am J Roentgenol. 2021 Jan;216(1):W5. doi: 10.2214/AJR.20.24538. Epub 2020 Oct 28.
The Role of Transthoracic Ultrasound in the novel Coronavirus Disease (COVID-19): A Reappraisal. Information and Disinformation: Is There Still Place for a Scientific Debate? Front Med. 2020;7:271. doi:10.3389/fmed.2020.00271
Interstitial Lung Diseases. In: Thoracic Ultrasound and Integrated Imaging. Cham: Springer International Publishing;2020:61-82. doi:10.1007/978-3-319-93055-8_5
The artificial count of artifacts for thoracic ultrasound: what is the clinical usefulness? J Clin MonitComput. 2020. doi:10.1007/s10877-020-00484-0
The Pathologic Patterns Detectable by Transthoracic Ultrasonography Are Only thePleural and Subpleural Ones and Are Not Specific: Why Compare Them With High-Resolution Computed Tomography? J Ultrasound Med.2018;37(7):1847-1848. doi:10.1002/jum.14510
Transthoracic ultrasound in the assessment of pleural andpulmonary diseases: use and limitations. Radiol Med. 2014;119(10):729-740. doi:10.1007/s11547-014-0385-0
Usefulness of lung ultrasound imaging in COVID-19 pneumonia: The persisting need of safety and evidences.Echocardiography. 2020 Jul;37(7):1138-1139. doi: 10.1111/echo.14769. Epub 2020 Jun 23.
Dogra V, Rubens DJ. Ultrasound Secrets. Hanley & Belfus; 2004.
Lung Ultrasound in COVID-19 Patients - More Shadows Than Information . Ultraschall in Med. 2020 Apr 15".Ultraschall Med. 2020 Aug;41(4):439-440. doi: 10.1055/a-1177-3156. Epub 2020 Jun 25.
Gas at abdominal US: Appearance, relevance, and analysis of artifacts. Radiology. 1999;210(1):113-123. doi:10.1148/radiology.210.1.r99ja12113
Response to pleuropulmonary US examination artifacts: '"errors in images."' Ultrasound Med Biol. 2010;36(2):356-357. doi:10.1016/j.ultrasmedbio.2009.08.015. Epub 2009 Nov 27
Lung ultrasonography for the diagnosis of 11 patients with acute respiratory distress syndrome due to bird flu H7N9 infection. Virol J. 2015;12(1):176. doi:10.1186/s12985-015-0406-1
Subclinical pulmonary congestion is prevalent in nephrotic syndrome. Kidney Int. 2016;89(2):421- 428. doi:10.1038/ki.2015.279
Characterization of the normal pulmonary surface and pneumonectomy space by reflected ultrasound. J Ultrasound. 2011;14(1):22-27. doi:10.1016/j.jus.2011.01.004
Diagnosis of coronavirus disease 2019 pneumonia in pregnant women: can we rely on lung ultrasound? Am J Obstet Gynecol. 2020 Oct;223(4):615. doi: 10.1016/j.ajog.2020.06.028. Epub 2020 Jun 15
Lung ultrasound imaging in avian influenza A (H7N9) respiratory failure. Crit Ultrasound J. 2014;6(1):1-8. doi:10.1186/2036-7902-6-6
The Role of Chest Imaging in Patient Management during the COVID-19 Pandemic: A Multinational Consensus Statement from the Fleischner Society. Radiology. April 2020:201365. doi:10.1148/radiol.2020201365
ACR Recommendations for the use of Chest Radiography and Computed Tomography (CT) for Suspected COVID-19 Infection | American College of Radiology. https://www.acr.org/Advocacy-and-Economics/ACR-Position-Statements/Recommendations-for-Chest-Radiography-and- CT-for-Suspected-COVID19-Infection. Accessed April 8, 2020.
Author Response
Dear Reviewer,
Thank you for your valuable comments and suggestions. We are completely agreeing with you and we have applied your suggestions that will increase and also enhance the quality of this pictorial review and we are very grateful to you. We have highlighted all the limits of LUS in the section: 4.1 Lung ultrasound: role and limitations, and we have added all the suggested references that you can find marked in red with the reference number 54 and references from 64-78. We have deleted all the references that were preprint and we have rearranged some references that you can find in red.
We report also the references suggested marked in red:
- Marino, F.; Martorano, C.; Tripepi, R.; Bellantoni, M.,;Tripepi, G.; Mallamaci, F.,;Zoccali, C. Subclinical pulmonary congestion is prevalent in nephrotic syndrome. Kidney Int 2016, 89, 421-428. DOI: 10.1038/ki.2015.279.
- Tsai; N. W.; Ngai, C. W.; Mok, K. L.; Tsung, J. W. Lung ultrasound imaging in avian influenza A (H7N9) respiratory failure. Ultrasound J.2014, 6, 1-8; DOI:10.1186/2036-7902-6-6.
- Sperandeo, M.; Rotondo, A.; Guglielmi, G.; Catalano, D.; Feragalli, B.; Trovato, G.M. Transthoracic ultrasound in the assessment of pleural and pulmonary diseases: use and limitations. Med. 2014, 119, 729-740; DOI: 10.1007/s11547-014-0385
- Sperandeo, M.; Varriale., A.; Sperandeo, G.; Bianco, M.R.; Piattelli, M.L.; Bizzarri, M. et al. Characterization of the normal pulmonary surface and pneumonectomy space by reflected ultrasound. Ultrasound 2011, 14, 22-27; DOI: 10.1016/j.jus.2011.01.004.
- Wilson, S.R.; Burns, P.N.; Wilkinson, L.M.; Simpson, D.H.; Muradali, D. Gas at abdominal US: appearance, relevance, and analysis of artifacts. Radiology 1999, 210, 113-123; DOI: 10.1148/radiology.210.1.r99ja12113
- Dogra, V.; Rubens, D.J. Ultrasound secrets; Hanley & Belfus: Philadelphia, 2004; pp. 8-13.
- Sperandeo, M.; Trovato, G. Lung Ultrasound in COVID-19 Patients - More Shadows Than Information - Letter to the Editor on the Article “W. LU et al. Ultraschall in Med. 2020 Apr 15”; Ultraschall Med. 2020, 41, 439-440; DOI: 10.1055/a-1177-3156.
- Trovato, G.M.; Sperandeo, M. Usefulness of lung ultrasound imaging in COVID-19 pneumonia: The persisting need of safety and evidences. Echocardiography 2020, 37, 1138-1139; DOI: 10.1111/echo.14769..
- Sperandeo, M.; Quarato, C.M.I.; Rea, G. Diagnosis of coronavirus disease 2019 pneumonia in pregnant women: can we rely on lung ultrasound? J. Obstet. Gynecol. 2020, 223, 615;DOI: 10.1016/j.ajog.2020.06.028.
- Tinti, M.G.; Rea, G.; Frongillo, E.; Saponara, A.; Sperandeo, M. The Pathologic Patterns Detectable by Transthoracic Ultrasonography Are Only the Pleural and Subpleural Ones and
Are Not Specific: Why Compare Them With High-Resolution Computed Tomography? J Ultrasound Med. 2018, 37, 1847-1848; DOI: 10.1002/jum
- Sperandeo, M.; Carnevale, V.; Varriale, A. Response to pleuropulmonary US examination artifacts: “errors in images”. Ultrasound Med. Biol. 2010, 36, 356-357; DOI:https://doi.org/10.1016/j.ultrasmedbio.2009.09.005
- Quarato, C.M.I.; Venuti, M.; Sperandeo, M. Diagnosis of Coronavirus Disease (COVID-19) Pneumonia: Is Lung Ultrasound the Better Choice? AJR Am. J. Roentgenol. 2021, 216, W5; DOI: 10.2214/AJR.20.24538.
- Quarato, C.M.I.; Venuti, M.; Lacedonia, D; Simeone, A.; Dimitri, L.M.C,; Rea, G.; Ferragalli, B.; Sperandeo, M. The Role of Transthoracic Ultrasound in the novel Coronavirus Disease (COVID-19): A Reappraisal. Information and Disinformation: Is There Still Place for a Scientific Debate? Med. 2020, 7, 271; DOI: 10.3389/fmed.2020.
- Sperandeo, M.; Rea G. Interstitial Lung Diseases. In Thoracic Ultrasound and Integrated Imaging; Feletti, F., Malta, B., Aliverti A., Eds; Springer: International Publishing, 2020, pp. 61-82; DOI: https://doi.org/10.1007/978-3-319-93055-8_
- Zhang, Y. K.; Li, J.; Yang, J. P.; Zhan, Y.; Chen, J. Lung ultrasonography for the diagnosis
of 11 patients with acute respiratory distress syndrome due to bird flu H7N9 infection. Virol J 2015, 12, 1-5; DOI: 10.1186/s12985-015-0406-1
- Quarato, C.M.I.; Venuti, M. & Sperandeo, M. The artificial count of artifacts for thoracic ultrasound: what is the clinical usefulness? J. Clin. Monit. Comput. 2020, 34, 1379–1381.DOI: doi.org/10.1007/s10877-020-00484-0.
We have also added in the section 4.2 Lung ultrasound: protocols the modality of execution of LUS, the frequency of the probes and all the parameters also on the base of the references that you have suggested. We have also added another reference in this section: the reference number 86: Buonsenso D; Piano A; Raffaelli F; Bonadia N; De Gaetano Donati K; Franceschi F. Point-of-care lung ultrasound findings in novel coronavirus disease-19 pnemoniae: A case report and potential applications during COVID-19 outbreak. Eur. Rev. Med. Pharmacol. Sci. 2020, 24:2776–2780.DOI: 10.26355/eurrev_202003_20549. We have added other images (Fig 3, Fig 4). In the Fig 3 we have described the Lines A and B artifacts with also an example, in the image c, of a pleural thickness and B artifacts in fibrosis to underline the low specificity of LUS. However, we are not expertise of LUS as we are emergency Radiologists that work on CT and the image b and c of the Figure 3 have been granted by the Head of the Ultrasound department of San Giuseppe Moscati, Luigi Monaco. We hope that these images satisfied you. In the Figure 4 we described an example of the anatomical division in 8 zones for the LUS execution.
This references were discussed in the text.
- Radiological Society of North America Expert Consensus Statement on Reporting Chest CT Findings Related to COVID-19. Endorsed by the Society of Thoracic Radiology, the American College of Radiology, and RSNA. Radiology: Cardiothoracic Imaging 2020;2(2). doi: 10.1148/ryct.2020200152
Radiological Society of North America Expert Consensus Statement on Reporting Chest CT Findings Related to COVID-19. Endorsed by the Society of Thoracic Radiology, the American College of Radiology, and RSNA J Thorac Imaging. 2020 Jul;35(4):219-227
ACR Recommendations for the use of Chest Radiography and Computed Tomography (CT) for Suspected COVID-19 Infection | American College of Radiology. https://www.acr.org/Advocacy-and-Economics/ACR-Position-Statements/Recommendations-for-Chest-Radiography-and- CT-for-Suspected-COVID19-Infection. Accessed April 8, 2020.
In particular, to underline the role of each modalities and the controversial role of LUS, because there isn’t a consensus of the International guidelines of the use of LUS in COVID-19 pneumonia diagnosis and its management we have added the Table 1 that summarized the main International guidelines. We have added other references (number 17,18, 20,21) because we found to summarized all of the International guidelines. You can find the new references marked in red and we have rearranged also in this section (Introduction) some references. All the rearranged references in the text were reported in red. The previous Table 1 becomes the Table 3 and summarized the general indications of each imaging modalities according to the literature. All the modifications in the text are written in red and also the arranged references are written in red. We have added in the text also a section on the histopathological correlations (section 6) on the base of the suggestion of another Reviewer and we have also added new references. Because we have added two LUS figures, we have edited the numeric orders of each figures in the text and also for the references, we have reorganized the numeric order in the text. The manuscript has been edited for the English Revision by the MDPI service for the English editing. We hope that you will find all the modifications satisfactory.
With Kind Regards.
Reviewer 3 Report
Dear Author,
the manuscript is a narrative review of the current diagnostic possibilities of the different imaging methods. However, the text needs a thorough stylistic revision.
Furthermore, the reading of the manuscript does not reach the set goal. I would have expected the authors to propose (through an analysis of the "recommendations" listed by the various radiology companies) a "guideline" in the approach to patients with suspected or confirmed COVID19 infection. More interesting would have been to read about a correlation between the different imaging data and patho-physio-morphological correlations. Therefore, it does not present the elements of originality. Aspects that, more specifically, deserve to be explained are:
1. The "Method" section contains overly superficial review criteria that prevent the repeatability of the search. Furthermore, the results of the selection made are missing.
2. Table 1 would have been more interesting if indications, advantages, or disadvantages had been related to the individual recommendations dictated by the respective radiology companies, rather than citing them for the CXR alone.
3. The "Structured report example" lacks an anamnestic section with the clinical-functional aspects, as well as a reference to a qualitative-quantitative classification and/or measurement that may be functional for the clinician receiving the examination (such as CO -RADS or other eventual proposed by the authors and tested in its case series).
Therefore, it is rejected.
Author Response
Dear Author,
the manuscript is a narrative review of the current diagnostic possibilities of the different imaging methods. However, the text needs a thorough stylistic revision.
Furthermore, the reading of the manuscript does not reach the set goal. I would have expected the authors to propose (through an analysis of the "recommendations" listed by the various radiology companies) a "guideline" in the approach to patients with suspected or confirmed COVID19 infection. More interesting would have been to read about a correlation between the different imaging data and patho-physio-morphological correlations. Therefore, it does not present the elements of originality. Aspects that, more specifically, deserve to be explained are:
- The "Method" section contains overly superficial review criteria that prevent the repeatability of the search. Furthermore, the results of the selection made are missing.
- Table 1 would have been more interesting if indications, advantages, or disadvantages had been related to the individual recommendations dictated by the respective radiology companies, rather than citing them for the CXR alone.
- The "Structured report example" lacks an anamnestic section with the clinical-functional aspects, as well as a reference to a qualitative-quantitative classification and/or measurement that may be functional for the clinician receiving the examination (such as CO -RADS or other eventual proposed by the authors and tested in its case series).
Therefore, it is rejected.
Dear Reviewer,
we are very sorry about your decision and we are not agreeing with it. This manuscript represents one of few Reviews that describes in details the role of each imaging modality, the limits and each scores used for COVID-19 pneumonia. However, on the base of your comments we have also added the Table 1 that describes the main International guidelines and also a section about the imaging and histopathological correlations. We have also highlighted the limits of LUS. The manuscript has been edited also in English and we have deleted the part of Material and Methods because it isn’t a Systematic Review. However, we believe that your suggestions have been increased the quality of this manuscript. In the structured report we have added also the CO-RADS and the example of the structured report was made on the base of ESTI and SIRM guidelines and it can be helpful for Radiologists. The structured report should contain general indications of the clinical information and it is used as a standardized language for Radiologists in case of a COVID-19 pneumonia findings.
With Kind Regards
Reviewer 4 Report
A comprehensive review, useful for the clinical practice.
Author Response
Dear Reviewer,
Thank you for your valuable comments and appreciations. We are grateful to you. We have also added some corrections in the text that you can find in red. Therefore, we have added the table 1 that summarized the main International guidelines and the previous table 1 becomes the table 3 that summarized the general guidelines on the base of the current literature. We have highlighted also the limits of the LUS, we have added the Figure 3 and 4 and we have added the section 6 on the histopathological correlations. So, the new references were marked in red and the rearranged references were written in red. All the modifications were written in red. The Manuscript has been also edited in English by the MDPI service for the English editing.We hope that you can also find all the modifications satisfactoryWith Kind regards
Round 2
Reviewer 2 Report
Dear authors,
I recommend the following minor changes:
194 small multifocal consolidations and dynamic air bronchograms in context, as well as pleural effusions and complications such as PNX
To be changed:
small multifocal consolidation adherent to subpleural surface, as well as pleural effusion.
Chest. 2016 May;149(5):1350-1. doi: 10.1016/j.chest.2016.02.684.
Ultrasound Diagnosis of Ventilator-Associated Pneumonia: A Not-So-Easy Issue.
no study or metaanalysis has so far demonstrated that such hyperechoic
images hyperechoic spots and/or bands, improperly called an air bronchograms do
really correspond to the CT imaging finding of air bronchogram. Instead,
we have shown that such images can also be detected in lung neoplasm masses
Transthoracic Ultrasound in Pneumothorax. Ann Thorac Surg. 2020 Jan;109(1):310. doi: 10.1016/j.athoracsur.2019.04.077. Epub 2019 Jun 15.
TUS cannot detect pneumothoraxes restricted to the mediastinal area . Moreover, the sliding sign could be physiologically absent at the lung apex, resulting in a false positive, namely for the presence of three ligaments (transverse-pleural, costal-pleural and vertebral-pleural). Many conditions, such as COPD, pulmonary fibrosis, emphysema, subcutaneous emphysema, that can raise a false positive, limiting even more the usefulness of the TUS. Anyway, also when a pneumothorax is suspected, further investigations are always required to determine its extent and depth
217 Compared with volumetric chest CT, LUS analyzes only 70% of the lungs, poorly
218 examines the posterior regions, which are the most affected in COVID-19 pneumonia, and
219 does not identify the central regions, surrounded by aerated lung [71,74,75]
to be changed :
Compared with volumetric chest CT, LUS explore , at best, with patient in a sitting position, only the 70% of the pleural surface : and does not identify the central regions , perhylar or subpleural not adherent to pleural surface , due to the interposition of also a very thin layer ( micron or mm) of air between the pleural surface surrounded by aerated lung.
245 The most commonly used protocols are the bedside lung ultrasonography in 246 emergency (BLUE),
To be change
The most commonly proposed protocols are the bedside lung ultrasonography in emergency (BLUE),
265 Some authors suggest using wireless probes and tablets, as they can be easily wrapped in plastic covers [50,86]. This sentence is irrelevant, I would not include it because the papers of the reference raises several elements of concerns for health professionals working in the field of Viral pneumonia and lung US.
Echocardiography. 2020 Jun 23 : 10.1111/echo.14769.doi: 10.1111/echo.14769
Author Response
Dear Reviewer,
Thank you for your valuable corrections and suggestions. The sentence in line 194 has been edited by your suggestions and we have added also this sentence: However, LUS cannot accurately detect the presence of air bronchogram as well as pneumothorax with the references numbers 57 and 58 that have been suggested by you and they are reported as the references number 57,58
Sperandeo, M.; Filabozzi P.; Carnevale, V. Ultrasound diagnosis of ventilator associated pneumonia: a not so easy issue. Chest 2016; 149:1350-1351.DOI: 10.1016/j.chest.2016.02.684.
Tinti, M.G..; Quarato C.M.I.; Sperandeo, M. Transtoracic ultrasound in pneumothorax. Ann Thorac Surg 2020; 109:310. doi: 10.1016/j.athoracsur.2019.04.077.
We have also edited the lines 217-219 as you have suggested in the following: Compared with volumetric chest CT, LUS explores, at best, with patient in a sitting position, only the 70% of the pleural surface and does not identify the central regions as well as the perihilar or subpleural regions that do not adherent to pleural surface, due to the interposition of also a very thin layer (micron or mm) of air between the pleural surface surrounded by aerated lung
The sentence in line 245 has been edited on the base of your suggestion: The most commonly proposed protocols are the bedside lung ultrasonography in emergency (BLUE).
We have also deleted the sentence in line 265 with the reference n 86 as you have suggested.
All the corrections that you have suggested are marked in black. However, we have added also a reference in the histopathological correlations n 169, because we have previously omitted the reference near this sentence. Most of the histopathological findings described in fatal cases of COVID-19 pneumonia are similar to those found in severe acute respiratory syndrome corona virus-1 (SARS-CoV-1), Middle East respiratory syndrome coronavirus (MERS-CoV) and also to virus Ebola [169] and we have added also the similarity with the virus Ebola that was already reported.
As the manuscript describes also the histopathological findings we have also added it in the conclusion with the sentence Most of the histopathological findings described in COVID-19 are similar to those found in severe acute respiratory syndrome corona virus-1 (SARS-CoV-1), Middle East respiratory syndrome coronavirus (MERS-CoV) and also to virus Ebola
Due to the fact the we have added the references number 57,58 on your suggestions and deleted the reference number 86 and added the reference number 169, we have edited the order of the other following reference that are reported in red
We hope that will find all corrections satisfactory
With Kind Regards
Reviewer 3 Report
Dear Author, the great revision work that you have carried out in responding to the observations that had been made is appreciable.
Author Response
Dear Reviewer,
Thank you for your favorable response and we are grateful and honored of your appreciation
With Kind regards